# Conformer: A Parallel Segmentation Network Combining Swin Transformer and Convolution Neutral Network

Yanbin Chen[1][0000−0003−0231−1390], Zhicheng Wu[2][0000−0002−8342−0507], Hao Chen[3][0000−0002−6849−9413], and Mingjing Yang[✉]

College of Physics and Information Engineering, Fuzhou University, Fuzhou, China
{yangmj5}@fzu.edu.cn

**Abstract.** Abdominal organ segmentation can help doctors to have a more intuitive observation of the abdominal organ structure and tissue lesion structure, thereby improving the accuracy of disease diagnosis. Accurate segmentation results can provide valuable information for clinical diagnosis and follow-up, such as organ size, location, boundary status, and spatial relationship of multiple organs. Manual labels are precious and difficult to obtain in medical segmentation, so the use of pseudo-labels is an irresistible trend. In this paper, we demonstrate that pseudo-labels are beneficial to enrich the learning samples and enhance the feature learning ability of the model for abdominal organs and tumors. In this paper, we propose a semi-supervised parallel segmentation model that simultaneously aggregates local and global information using parallel modules of CNNS and transformers at high scales. The two-stage strategy and lightweight network make our model extremely efficient. Our method achieved an average DSC score of 89.58% and 3.78% for the organs and tumors, respectively, on the testing set. The average NSD scores were 93.33% and 1.81% for the organs and tumors, respectively. The average running time and area under GPU memory-time curve are 14.85s and 15962MB.

**Keywords:** Abdominal organ and tumor segmentation · Hybrid architecture · Pseudo-label.

## 1 Introduction

The CT scan is a standard diagnostic method for abdominal-related diseases in clinical practice. Through CT scans, doctors can obtain a more intuitive observation of the abdominal organ structures and pathological changes, thereby improving the accuracy of disease diagnosis. Accurate segmentation results can provide valuable information for clinical diagnosis and follow-up, such as organ size, lesion position, boundary status, and spatial relationships of multiple organs[20]. In clinical practice, doctors often have to manually annotate organ segmentation, which is both time-consuming and prone to subjective opinions.

Developing an automated multi-organ segmentation model using deep learning can improve the efficiency of clinical workflows, including disease diagnosis, prognosis analysis, and treatment planning.

It is known from previous work [12,11,2]that the combination of Transformer and CNN has achieved remarkable results in the field of multi-object segmentation. Our proposed model leverages the capabilities of Convolutional Neural Networks (CNN) in conjunction with the state-of-the-art **Swin Transformer**[13] to employ parallel modules at deep stages. These modules are adept at concurrently aggregating both local and global information, thereby enabling more effective segmentation. To optimize computational efficiency, we employ standard convolutional blocks exclusively in the shallow layers of our network, mitigating excessive computational demands. In addition, our network adopts a two-stage cascade framework, with the first stage for organ region localization and the second stage for whole-organ and tumor-refined segmentation. We believe that simultaneous learning of tumors and organs facilitates the acquisition of tumor information. To further augment our dataset and bolster tumor labeling, we employ a trained fine organ segmentation model to generate pseudo-labels for organs from data that is only annotated for tumors. This strategy not only expands the available tumor data but also leverages the interplay between organ and tumor structures, leading to more comprehensive and accurate segmentation results.

The main contributions of this work are summarized as follows:

(1)Proposing a Parallel CNN and Transformer hybrid modules for information aggregation which effectively aggregates local and global information.

(2)The two-stage cascade framework from coarse to fine can effectively reduce the redundancy of image information and alleviate the computational load.

## 2   Method

We propose a two-stage strategy as shown in Figure 1. In the first stage, coarse segmentation is performed on the binary classification to obtain the region of interest (ROI) for the entire organ. In the second stage, fine segmentation is performed on the cropped ROI.

### 2.1   Preprocessing

All the training images are uniformly preprocessed as follows:

- Referring to Liu et al.[12], all the data were resampled to the same size. The image data and the segmentation were interpolated using bi-cubic interpolation and nearest neighbor methods. The images used for coarse segmentation training in the first stage were resampled to $64 \times 64 \times 64$ size, and the images used for fine segmentation training in the second stage were resampled to $96 \times 192 \times 192$ size.
- The data was standardized by Z-Score.

## 2.2   Proposed Method

Figure 2 shows the encoder-decoder architecture of our model. There are four stages in this configuration. In the first two stages, convolutional blocks are used, while the subsequent two stages employ parallel blocks. Within the first two scales, the skip-connection linking the pure convolutional blocks is implemented using a concatenation operation. In the last two stages, the skip-connection that connects the convolutional blocks uses the same concatenation operation, while the addition operation is applied between the transformers. The two-stage strategy and the utilization of parallel hybrid convolution at the deep stages effectively reduce the complexity of the model while still maintaining segmentation accuracy.

We employ two models in the fine segmentation part to correspond to organ segmentation and tumor segmentation. Both models have the same network framework. During training, we utilize the summation of Dice loss and cross-entropy loss as the loss function. When it comes to training the organ model, a noteworthy concern arises when using partial labels directly from the dataset. This approach has the potential to cause confusion within the organ model, resulting in incomplete organ segmentation outcomes. Specifically, this manifests as the segmentation of fewer than the expected 13 organ classes. To mitigate this issue, we adopt a different strategy. We utilize both the complete set of organ labels and an additional 1800 pseudo labels generated through the FLARE22 winning algorithm [8] and the best accuracy algorithm [21] to facilitate the training of the organ model. Subsequently, we employ the trained organ model to generate pseudo labels specifically for unlabeled organs when dealing with partially labeled data. These pseudo-labels are then integrated into the training process of the tumor model. The resulting dataset, which consists of a combination of labeled and pseudo-labeled data, is used to train the tumor model. Importantly, the predictions generated by the tumor model are limited to only retaining the tumor segmentation results. Notably, our empirical observations indicate improved performance when training both the tumor and organ models simultaneously.

**Parallel Block**  The Conv Block consists of $3 \times 3 \times 3$ conv layer, IN layer and GELU layer. The composition of Swin Block 3D[13] is shown in Figure 3. It consists of two basic units, W-MSA and SW-MSA. The former computes the similarity between tokens in the same window. This segmentation strategy is helpful to deal with large-size images, so that the model can effectively deal with large-scale data. The latter moves the input token in each dimension by s units, and then calculates the similarity within the window, which is beneficial to capture the information between different blocks and obtain global image information. The first unit consists of a norm layer(LN) , a Window Multi-head Self-attention 3D(W-MSA-3D) module, a norm layer, and an MLP module, in order. The second unit uses the Window Shifting Multi-head Self-Attention 3D (SW-MSA-3D) module to replace the W-MSA-3D module in the first unit, and the rest of the structure is the same as the first unit. We need to convert

voxels to tokens before performing Swin Block, flattening a block of voxels of dimension $[H \times W \times D]$ into a one-dimentional token of length $H \times W \times D$ by matrix dimension transformation. Similarly, after the self-attention calculation is completed, the tokens are converted into voxels.

**Loss function** We utilize a combination of the Dice loss and cross-entropy loss. The overall loss of our Conformer is defined as:

$$L_S = L_{WCE}(G, S) + L_{dice}(G, S), \tag{1}$$

$L_{WCE}$ is formulated as:

$$L_{WCE}(G, S) = \frac{\sum_{c \in C} \| - w_c \cdot S_c \cdot log(G_c)\|}{H \cdot W \cdot D}, \tag{2}$$

where $\|\cdot\|$ denotes the L1 norm, $w_c$ is the weight for c-th class. $L_{dice}$ is formulated as:

$$L_{dice}(G, S) = 1 - \sum_{c \in C} \frac{2 \sum_{i=1}^{N} G_c^i S_c^i}{\sum_{i=1}^{N} G_c^i G_c^i + S_c^i S_c^i}, \tag{3}$$

where $G_c^i$ , $S_c^i$ respectively denote the ground truth and output of voxel i for class c.

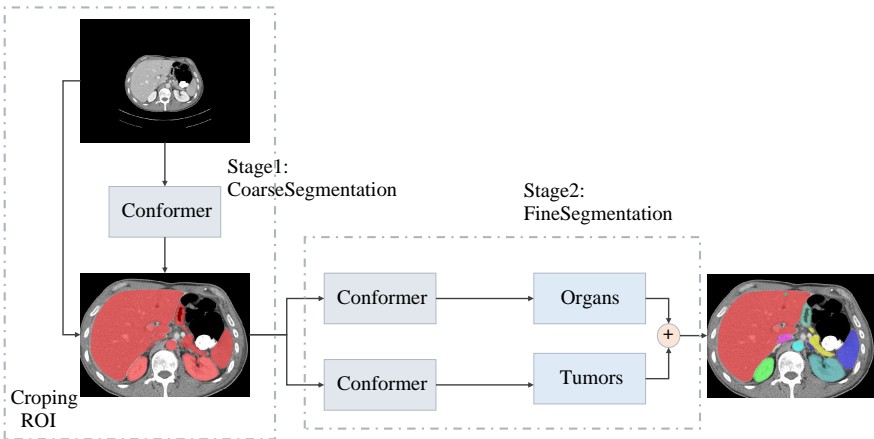

**Fig. 1.** Overview of our two-stage segmentation frame

## 2.3   Post-processing

In post-processing, the largest connected component is employed to refine the segmentation results. It is noteworthy that segmentation algorithms occasionally generate diminutive, extraneous segmented regions, which can be attributed

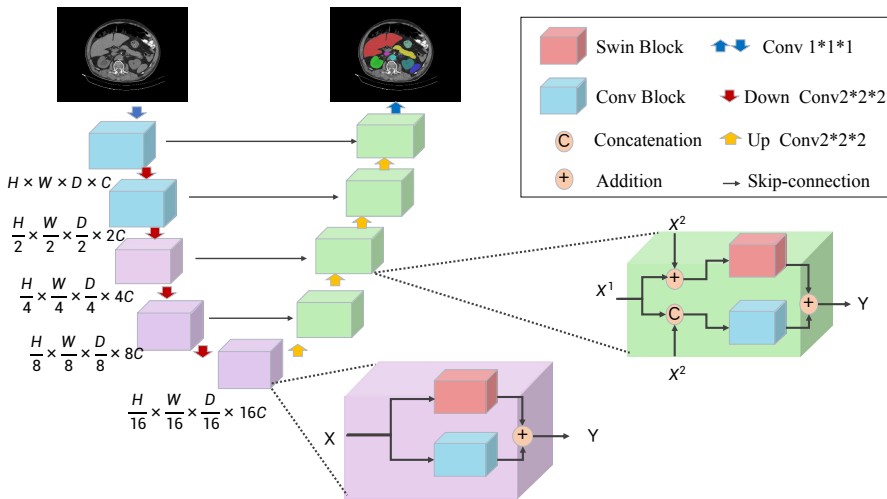

**Fig. 2.** Network architecture: Our network framework follows the encoder-decoder architecture, using Conv Blocks in the shallow layers and Parallel Blocks consisting of Convolutional Blocks and Swin Block in the deep layers.

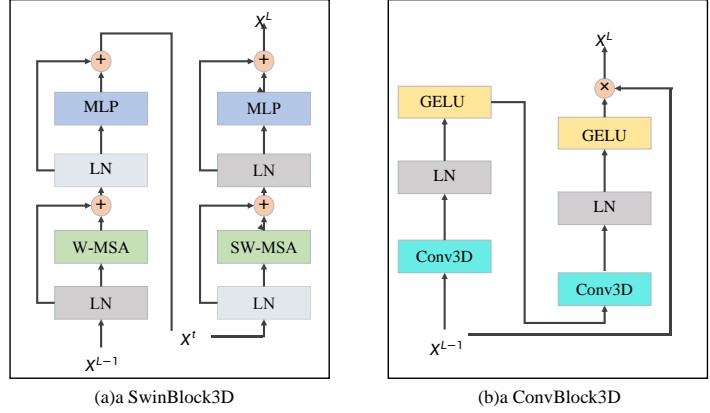

(a)a SwinBlock3D                    (b)a ConvBlock3D

**Fig. 3.** Overview of the structure of Swin Block and Conv Block.

to inherent noise or algorithmic inaccuracies. Through the application of the largest connected component, the extraneous segments can be effectively discarded, thereby mitigating the likelihood of incorrect segmentation outcomes. Additionally, certain instances may arise where segmentation algorithms partition a single anatomical organ into multiple disjointed segments. By utilizing the concept of the largest connected component, fragmented segments can be merged into a continuous anatomical region. This process improves the comprehensibility and visual consistency of the segmentation results.

## 3  Experiments

### 3.1  Dataset and evaluation measures

The FLARE 2023 challenge is an extension of the FLARE 2021-2022 [15][16], aiming to aim to promote the development of foundation models in abdominal disease analysis. The segmentation targets cover 13 organs and various abdominal lesions. The training dataset is curated from more than 30 medical centers under the license permission, including TCIA [3], LiTS [1], MSD [19], KiTS [6,7], autoPET [5,4], TotalSegmentator [22], and AbdomenCT-1K [17]. The training set includes 4000 abdomen CT scans where 2200 CT scans with partial labels and 1800 CT scans without labels. The validation and testing sets include 100 and 400 CT scans, respectively, which cover various abdominal cancer types, such as liver cancer, kidney cancer, pancreas cancer, colon cancer, gastric cancer, and so on. The organ annotation process used ITK-SNAP [23], nnU-Net [10], and MedSAM [14].

The evaluation metrics encompass two accuracy measures—Dice Similarity Coefficient (DSC) and Normalized Surface Dice (NSD)—alongside two efficiency measures—running time and area under the GPU memory-time curve. These metrics collectively contribute to the ranking computation. Furthermore, the running time and GPU memory consumption are considered within tolerances of 15 seconds and 4 GB, respectively.

### 3.2  Implementation details

**Environment settings**  The development environments and requirements are presented in Table 1.

**Training protocols**  We applied two models in the fine segmentation part to correspond to organ segmentation and tumor segmentation. We divided the data provided by FLARE 2023 into four types: (1) The cases without tumors but with whole organ labels. (2) The cases with partial organ labels, some of which included tumor labels. (3) The cases with only tumor labels. (4) The 1800 organ pseudo-label without tumors labels generated by the top teams last year[9]. In the training phase, the first type of data is used to train the coarse segmentation model, while the first and fourth types of data are used to train the organ fine

**Table 1.** Development environments and requirements.

| | |
|---|---|
| System | Ubuntu 20.04.4 LTS |
| CPU | 12th Gen Intel(R) Core(TM) i9-12900K CPU@3.20GHz |
| RAM | 42GB |
| GPU (number and type) | One GeForce RTX 3080 12GB |
| CUDA version | 12.0 |
| Programming language | Python 3.8 |
| Deep learning framework | torch 1.12.1, torchvision 0.13.1 |
| Specific dependencies | |
| Code | |

segmentation model. When training the tumor model, we discovered that only the fourth data label had a minimal impact. When the third type of data was utilized, the effect was enhanced. We convinced that learning about both organs and tumors simultaneously can enhance the understanding of tumor information. We applied the trained fine organ segmentation model to the organ pseudo-labels generated from the third type of data to obtain labels that encompass entire organs and tumors. We therefore adopted the new third data to train the tumor model, setting it to 15 classifications, but leaving only the final tumor segmentation output.

We apply the following augmentation methods to the training data: random rotation, scaling, addition of Gaussian white noise, Gaussian blur, gamma transformation, and elastic deformation. Each Swin Block contains a multi-head attention mechanism layer. According to the design of **Swin Unet**[2], the number of multi-head attention mechanisms in the encoder is 3, 6, and 9, respectively. Similarly, the number of multi-head attention mechanisms used in the decoder is 6,3 respectively. In the two-stage cascade network, the input data for coarse segmentation is $64 \times 64 \times 64$, and its attention window size is set to $4 \times 4 \times 4$. On the other hand, the input for fine segmentation is $96 \times 192 \times 192$, and the attention window size is set to $4 \times 3 \times 3$. The base number of channels is set to 16. Batch size is set to 1.

## 4 Results and discussion

### 4.1 Quantitative results on validation set

The quantitative results of the validation set are shown in Table 3. Our method achieves an average DSC of 83.88% $\pm$ 11.12% and an average NSD of 87.50% $\pm$ 11.00%. Table 7 shows the effectiveness of utilizing unlabeled data. Controlled experiments were performed using the same training configuration described in Section 3.2. Compared to using only labeled data to train the organ segmentation model, the utilization of pseudo-labeled data significantly enhanced the accuracy of organ segmentation. The average organ DSC score increased from 74.75% to 89.12%, and the organ NSD increased from 80.71% to 93.18%. We attempted to

**Table 2.** Training protocols for coarse model.

| | |
|---|---|
| Network initialization | |
| Batch size | 1 |
| Patch size | 64×64×64 |
| Total epochs | 500 |
| Optimizer | AdamW |
| Initial learning rate (lr) | 0.01 |
| Lr decay schedule | Cosine Annealing LR |
| Training time | 3.05 hours |
| Loss function | Dice + Cross entropy |
| Number of model parameters 6.66M[1] | |
| Number of flops | 20146.29M[2] |

**Table 3.** Training protocols for the refine model.

| | |
|---|---|
| Network initialization | organ/tumor |
| Batch size | 1 |
| Patch size | 96×192×192 |
| Total epochs | 500 |
| Optimizer | AdamW |
| Initial learning rate (lr) | 0.01 |
| Lr decay schedule | Cosine Annealing LR |
| Training time | 21.43 hours/23.56 |
| Loss function | Dice + Cross entropy |
| Number of model parameters 6.66M[3] | |
| Number of flops | 272606.26M/272662.88[4] |

**Table 4.** Segmentation DSC of abdominal organs and tumors.

| Target | Public Validation | | Online Validation | | Testing | |
|---|---|---|---|---|---|---|
| | DSC(%) | NSD(%) | DSC(%) | NSD(%) | DSC(%) | NSD (%) |
| Liver | 98.03 ± 1.02 | 98.63 ± 1.83 | 97.73 | 98.18 | 96.18 | 96.77 |
| Right Kidney | 93.20 ± 15.10 | 92.28 ± 15.46 | 92.34 | 91.19 | 94.24 | 92.84 |
| Spleen | 97.24 ± 3.43 | 97.23 ± 6.42 | 96.70 | 96.88 | 96.12 | 96.81 |
| Pancreas | 85.37 ± 7.60 | 95.31 ± 7.18 | 82.06 | 92.18 | 85.22 | 93.29 |
| Aorta | 96.99 ± 1.86 | 98.86 ± 2.62 | 97.05 | 98.90 | 97.71 | 99.20 |
| Inferior vena cava | 91.89 ±6.66 | 92.74 ±7.00 | 91.29 | 98.90 | 91.32 | 92.12 |
| Right adrenal gland | 85.28 ± 13.59 | 94.76 ± 14.10 | 85.14 | 98.90 | 86.66 | 95.59 |
| Left adrenal gland | 84.88 ± 7.52 | 95.02 ± 5.34 | 83.65 | 93.15 | 85.48 | 93.96 |
| Gallbladder | 80.29 ±28.45 | 79.95 ±29.45 | 81.41 | 81.59 | 79.70 | 81.26 |
| Esophagus | 82.65 ± 15.69 | 91.55 ± 15.94 | 84.00 | 93.16 | 89.58 | 97.49 |
| Stomach | 92.94 ± 6.67 | 95.69 ± 7.13 | 94.91 | 92.59 | 91.17 | 93.28 |
| Duodenum | 82.53 ± 8.05 | 94.13 ± 6.15 | 92.59 | 92.79 | 80.49 | 90.08 |
| Left kidney | 92.28 ± 14.28 | 91.10 ± 16.08 | 82.03 | 92.79 | 92.92 | 92.08 |
| Tumor | 10.63 ±25.72 | 7.81 ± 1.34 | 8.91 | 5.51 | 3.78 | 1.81 |
| Average | 83.88 ± 11.12 | 87.50 ± 11.00 | 89.12 | 93.18 | 83.45 | 86.79 |

exclusively utilize the tumor label data for training the tumor model as a binary classification, but the results were nearly negligible. When we used the data with both organ and tumor labels for training, we observed an improvement in the effectiveness of the tumor. Therefore, we employed the trained organ segmentation model to generate pseudo-organ labels. Subsequently, the data containing both organ and tumor labels was used to train the tumor model. The results showed an increase in both tumor metrics. Tumor NSD is 5.51% and Tumor DSC is 8.91%.

**Table 5.** Quantitative evaluation of segmentation efficiency in terms of the running them and GPU memory consumption.

| Case ID | Image Size | Running Time (s) | Max GPU (MB) | Total GPU (MB) |
|---|---|---|---|---|
| 0001 | (512, 512, 55) | 12.87 | 2778 | 13249 |
| 0051 | (512, 512, 100) | 12.82 | 2772 | 13652 |
| 0017 | (512, 512, 150) | 13.81 | 2778 | 14127 |
| 0019 | (512, 512, 215) | 17.44 | 2660 | 19189 |
| 0099 | (512, 512, 334) | 17.07 | 2660 | 18226 |
| 0063 | (512, 512, 448) | 21.63 | 2660 | 22592 |
| 0048 | (512, 512, 499) | 22.03 | 2772 | 23147 |
| 0029 | (512, 512, 554) | 27.84 | 3018 | 30459 |

**Table 6.**  Ablation study on the architecture.

| Method | Params (M) | FLOPs(G) |
|---|---|---|
| Conformer -1,2,3,4 | 7.35 | 353.78 |
| Conformer -2,3,4 | 6.79 | 285.28 |
| Conformer -3,4 | 6.66 | 266.21 |

**Table 7.** The effect of using unlabeled data.

| Target | Without unlabeled | | With unlabeled | |
|---|---|---|---|---|
| | DSC(%) | NSD(%) | DSC(%) | NSD(%) |
| Liver | 94.45 | 94.98 | 97.73 | 98.18 |
| Right Kidney | 88.06 | 89.44 | 92.34 | 91.19 |
| Spleen | 58.76 | 59.63 | 96.70 | 96.88 |
| Pancreas | 67.00 | 79.16 | 82.06 | 92.18 |
| Aorta | 89.51 | 90.95 | 97.05 | 98.90 |
| Inferior vena cava | 85.58 | 85.52 | 91.29 | 98.90 |
| Right adrenal gland | 74.45 | 90.54 | 85.14 | 98.90 |
| Left adrenal gland | 63.36 | 78.31 | 83.65 | 93.15 |
| Gallbladder | 67.28 | 67.44 | 81.41 | 81.59 |
| Esophagus | 70.47 | 80.47 | 84.00 | 93.16 |
| Stomach | 78.90 | 82.29 | 94.91 | 92.59 |
| Duodenum | 67.51 | 81.97 | 92.59 | 92.79 |
| Left kidney | 66.43 | 68.48 | 82.03 | 92.79 |
| Tumor | 6.58 | 3.44 | 8.91 | 5.51 |
| Average | 74.75 | 80.71 | 89.12 | 93.18 |

## 4.2    Qualitative results on validation set

Figure 4 shows our visualization results. Specifically, we present three examples of effective segmentation and three examples of poor segmentation. In cases where segmentation is successful, the results obtained closely align with the ground truth labels, demonstrating a high level of accuracy in our segmentation method. However, in FLARE23Ts 11 subject, the segmentation of the duodenum is notably poor. This discrepancy can be attributed to the presence of substantial deformations and blurred boundaries within the duodenum region. Furthermore, in FLARE23Ts 28 subject, our model incorrectly identifies the tumor as a part of the stomach, indicating a misclassification issue. Additionally, in FLARE23Ts 48 subject, the tumor is not successfully identified, highlighting the model's limitations in addressing certain tumor characteristics. This poor performance can be attributed to various factors. Firstly, the non-uniform sizes of tumors and their distribution across multiple organs pose a significant challenge to our segmentation model. Additionally, the limited availability of tumor samples hinders the model's learning capabilities. Therefore, enhancing the dataset with alternative methods to strengthen the representation of tumor samples may be a crucial approach for improvement.

## 4.3    Segmentation efficiency results on validation set

The segmentation efficiency is shown in Table 5. We compared the number of parameters and FLOPs using hybrid modules at different stages in Table 6. "-3,4" represents the use of hybrid modules in the third and fourth layers, and pure convolutional modules in the remaining layers. "-2.3,4" represents the utilization of hybrid modules in the second, third, and fourth layers. "-1,2.3,4" represents the utilization of hybrid modules at each stage. The proposed method is significantly faster than other methods in terms of inference time. This is due to our two-stage strategy and the use of Conv Blocks at shallow layers and Parallel Blocks exclusively at deep layers.

## 4.4    Results on final testing set

The effects of the test set evaluated by the official are shown in Table 8.

**Table 8.** The testing results from the official evaluation.

| Organ DSC | Organ NSD | Lesion DSC | Lesion NSD | Time | GPU Memory |
|-----------|-----------|------------|------------|--------|------------|
| 89.58%    | 93.33%    | 3.78%      | 1.81%      | 14.85s | 15962MB    |

## 4.5    Limitation and future work

The current state of tumor segmentation is characterized by a notable degree of inefficiency. To improve the accuracy of tumor segmentation, it is crucial

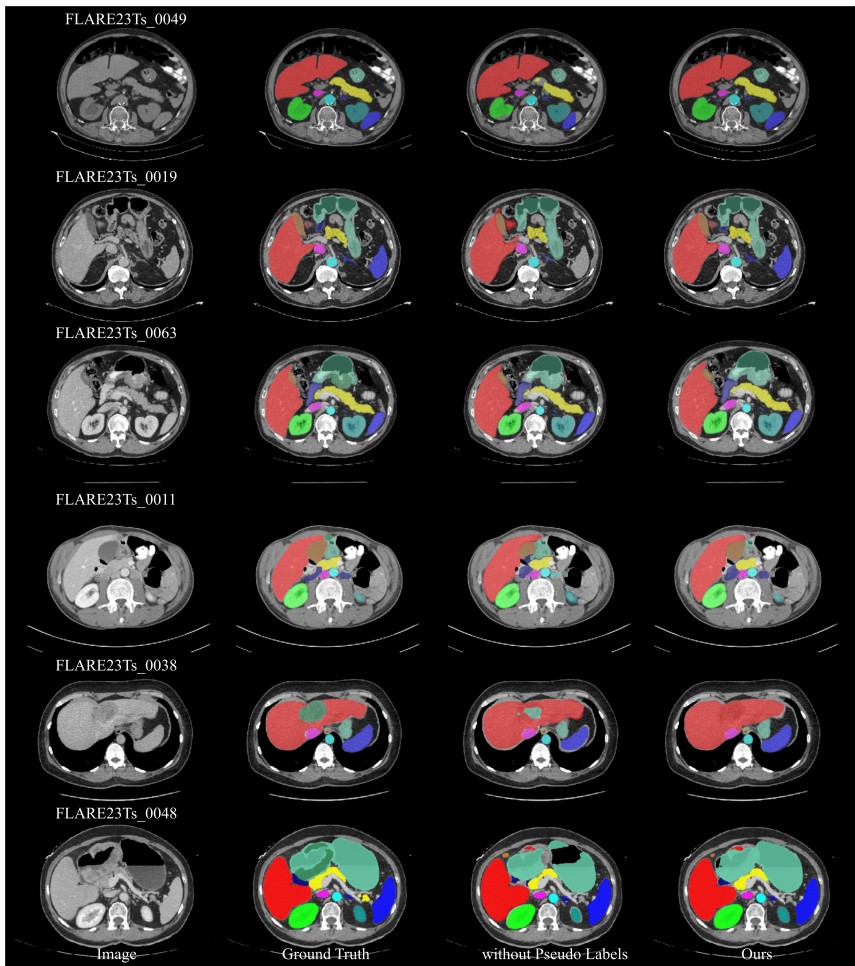

**Fig. 4.** Visualization of segmentation results of abdominal organs and tumors.

to develop strategies that focus on improving the quality of tumor samples. Notably, there is inherent heterogeneity in the tumor features exhibited across different anatomical organs. It is conceivable that one potential approach to addressing this heterogeneity involves dividing tumor segmentation into distinct segments that are specific to each organ. This approach seeks to acquire a refined understanding of the tumor characteristics associated with each organ. However, it is imperative to acknowledge that this level of detail inherently increases the computational workload associated with the segmentation process.

An additional strategy under consideration involves utilizing prior knowledge of the relative positions of organs. Such an approach is intended to reduce the occurrence of incorrect segmentation. Nevertheless, the variability observed in the relative positions of organs across different cases presents a significant challenge in determining a fair and appropriate range of positions.

## 5    Conclusion

We propose a two-stage segmentation network for the FLARE 2023 abdominal organ segmentation task. The network combining CNN and **Swin Transformer** effectively aggregates local features and global information. The use of pseudo-labels effectively enhances the accuracy of organ and tumor segmentation. In addition, our two-stage and lightweight network framework achieves high efficiency. Our method achieves an average organ DSC of 89.58% and an average organ NSD of 93.33%. In our environment configuration, the average running time for each example during testing cases is 14.85s, the average maximum GPU memory is 15962 MB.

**Acknowledgements** The authors of this paper declare that the segmentation method they implemented for participation in the FLARE 2023 challenge has not used any pre-trained models nor additional datasets other than those provided by the organizers. The proposed solution is fully automatic without any manual intervention. We thank all the data owners for making the CT scans publicly available and CodaLab [18] for hosting the challenge platform.

This work was supported by National Natural Science Foundation of China (62271149), Fujian Provincial Natural Science Foundation project(2021J02019, 2021J01578).

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

**Table 9.** Checklist Table. Please fill out this checklist table in the answer column.

| Requirements | Answer |
| --- | --- |
| A meaningful title | Yes |
| The number of authors ($\leq 6$) | 4 |
| Author affiliations and ORCID | Yes |
| Corresponding author email is presented | Yes |
| Validation scores are presented in the abstract | Yes |
| Introduction includes at least three parts: background, related work, and motivation | Yes |
| A pipeline/network figure is provided | Figure 1,2 |
| Pre-processing | Page 2 |
| Strategies to use the partial label | Page 3 |
| Strategies to use the unlabeled images. | Page 3 |
| Strategies to improve model inference | Page 3,6 |
| Post-processing | Page 3 |
| Dataset and evaluation metric section is presented | Page 5 |
| Environment setting table is provided | Table 6 |
| Training protocol table is provided | Table 2,3 |
| Ablation study | Page 8 |
| Efficiency evaluation results are provided | Table 5 |
| Visualized segmentation example is provided | Figure 4 |
| Limitation and future work are presented | Yes |
| Reference format is consistent. | Yes |