# OpenReview forum: "Conformer: A Parallel Segmentation Network Combining Swin Transformer and Convolution Neutral Network"
_MICCAI.org/2023/FLARE — Submitted to FLARE 2023_

### Official Review · Reviewer_Az8j · 2023-09-27
**A Parallel Segmentation Network Combining Swin Transformer and Convolution**

**Rating:** 6
**Confidence:** 5

**Review:**

The paper proposes a hybrid network that combines convolution and swin-transformer block for medical image segmentation.  The author also proposed a two stage coarse to fine segmentation pipeline to reduce the redundancy.
The paper is clear but the writing needs to be improved (check the grammar and typos).
E.g. "We employ two models in the fine segmentation part to correspond to organ 13 organ segmentation" has an additional "organ".
Also in the abstract, " the average maximum GPU memory was 279490MB", which is over 200GB of memory.

---

### Official Review · Reviewer_knCS · 2023-09-29
**Conformer: A Parallel Segmentation Network Combining Swin Transformer and Convolution Neutral Network**

**Rating:** 5
**Confidence:** 4

**Review:**

Strengths:

The approach proposes a semi-supervised parallel segmentation model that aggregates both local and global information using parallel modules of CNNS and transformers at high scales.


Weaknesses:

1，The authors should double-check the format of the paper, while there are several grammatical problems in the paper.

2， The methodology chapter, in particular, lacks a description of the loss function used for training and its corresponding formula.

3， The paper does not introduce strategies to enhance inference speed or reduce resource consumption.

4， The paper should be standardized for the expression of DSC coefficients, using percentages in some places and decimals in others.

5， Figure 4 includes two online validation datasets, but it lacks the public validation data, and the standard deviation is also missing.

---

### Official Review · Reviewer_15Yj · 2023-10-03
**Conformer: A Parallel Segmentation Network Combining Swin Transformer and Convolution Neutral Network**

**Rating:** 10
**Confidence:** 4

**Review:**

good

---

### Official Review · Reviewer_dhYt · 2023-10-04
**Conformer: A Parallel Segmentation Network Combining Swin Transformer and Convolution Neutral Network**

**Rating:** 8
**Confidence:** 5

**Review:**

This paper proposes an architecture that uses convolutional layers in the early stage and a hybrid mix of convolutional and attention (Swin) layers in the later stages. It also uses two different sets of branches (decoders) for organ and tumor segmentation. Furthermore, they also develop a coarse-to-fine segmentation approach.
This work builds up on the previously established and effective hybrid convolutional-attention approach for medical imaging, however, does leave a bunch of questions (that can be addressed in a later publication):
1. Why was the 4 stage network architecture split as 2-convolution and 2-hybrid? Some previous approaches (Attention Augmented convolutions for example) have tested the best approach for the split configuration.
2. In the event of splitting the branches - organ and tumor, there needs to be a little loss adjustment that accounts for tumor in a particular organ region that could lead to an increase in the tumor DSC scores.

Other than that, very well written paper - easy to understand, but needs a bit of revising on the grammatical front.

---

### Decision · Program_Chairs · 2023-10-24

Accept